# State-Level Prevalence and Associates of Opioid Dependence in the USA

**DOI:** 10.3390/ijerph19073825

**Published:** 2022-03-23

**Authors:** Janni Leung, Gary C. K. Chan, Samuel X. Tan, Caitlin McClure-Thomas, Louisa Degenhardt, Wayne Hall

**Affiliations:** 1National Centre for Youth Substance Use Research, The University of Queensland, Brisbane 4067, Australia; c.chan4@uq.edu.au (G.C.K.C.); samuel.tan@uq.net.au (S.X.T.); caitlin.mcclurethomas@uq.net.au (C.M.-T.); w.hall@uq.edu.au (W.H.); 2National Drug & Alcohol Research Centre, University of New South Wales, Sydney 2031, Australia; l.degenhardt@unsw.edu.au; 3Policy and Epidemiology Group, Queensland Centre for Mental Health Research, Brisbane 4076, Australia; 4Institute of Health Metrics and Evaluation, University of Washington, Seattle, WA 98195, USA

**Keywords:** opioid-related disorders, heroin dependence, analgesics, opioid, public health, socio-economic, socio-demographic

## Abstract

Traditionally, opioid-related disease burden was primarily due to heroin use. However, increases in extra-medical (or non-medicinal use of prescription opioids; NMPOs) use has precipitated the current overdose epidemic in North America. We aim to examine the state-level prevalence of heroin and NMPO dependence and their associations with opioid-related mortality and state-level socio-demographic profiles. Data were pooled from the 2005–2014 National Survey on Drug Use and Health (NSDUH). We examine opioid-related mortality from CDC WONDER (Cause of Death database) by the past year prevalence of DSM-IV heroin and NMPO dependence, by age and sex, and their associations with state-level socio-demographic characteristics from census data. State-level rates of heroin dependence were associated with opioid-related death rates in young and mid-aged adults, while rates of NMPO dependence were associated with opioid-related death rates across all ages. The prevalence of heroin dependence was positively associated with state-level GDP/capita and urbanity. State-level NMPO dependence prevalence was associated with higher unemployment, lower GDP/capita, and a lower high-school completion rate. The prevalence of heroin and NMPO dependence are associated with a broad range of geographical and socio-demographic groups. Taking a wider view of populations affected by the opioid epidemic, inclusive interventions for all are needed to reduce opioid-related disease burden.

## 1. Introduction

Opioid dependence and related harms in the United States (U.S.) represents a substantial public health issue [1]. There has been a five-fold increase in opioid-associated deaths since 1999 [2], with opioids implicated in two-thirds of all drug-overdose fatalities [2]. A recent study estimated that opioid overdoses accounted for 1.7 million years of life lost in 2016 [3]. Key drivers of opioid burden are heroin use and the non-medical use (also referred to as extra-medical use) of prescription opioids (NMPOs) [4]. Throughout most of the 20th century, the population health burden caused by opioids was mainly driven by heroin, but over the past two decades, the use, misuse, and dependence of NMPOs have become major public health concerns.

In the late 1990s and early 2000s, NMPO-associated mortality rose with the wave of increased opioid prescribing [5]. Pharmaceutical pain relievers, such as OxyContin and methadone, have been described as “gateway drugs” [6,7], with documented transitions from medical to non-medical use [8,9]. In the early 2010s, NMPO-related mortality plateaued after the implementation of public health strategies to address overprescribing [4], including restrictive legislation [10] and treatment with abuse-deterrent formulations [11]. However, in what is commonly referred to as the “second wave” of the opioid epidemic, heroin-related mortality then rose sharply, reflecting a transitional period from NMPO to heroin use [12]. From the mid-2010s onwards, there has been a “third wave” caused by a rapid increase in opioid-related mortality driven by illicitly manufactured fentanyl, a highly potent synthetic opioid [13].

The rates of opioid mortality vary by location, with the Northeast and Appalachian regions being the most heavily impacted [14,15]. The variability has been attributed to state-level differences, such as in prescribing attitudes [16], opioid regulation (e.g., prescription monitoring programs) [4], and local drug availability [17]. Opioid use is also associated with several individual characteristics. It is more prevalent in younger adults [18], with accompanying consequences: 12.4% of deaths in the 15–24 years age group were opioid-related in 2016 [3]. Other factors linked to opioid use include male sex [4,19], fewer years of educational attainment [20], and rurality of residence [4]. There is a growing literature around the effect of unemployment, rurality, and macro-economic conditions on opioid-related mortality [21].

There has been a number of previous state-specific studies on the epidemiology of opioid use in the U.S., including studies in Ohio [22], Connecticut [23], and New York [24]. A gap in the literature is the limited research that includes data on all states to examine state-level factors associated with opioid-related harms. This study examines the prevalence of NMPO and heroin dependence by mortality and socio-demographic associates of the states to identify population risk factors on a state-level.

## 2. Materials and Methods

### 2.1. Study Design

This is a cross-sectional analysis of state-level prevalence, mortality, and socio-demographic variables between 2005–2014. Three data sources were used for this study with information presented in the corresponding sections below. The state-level prevalence estimates were sourced from the National Survey on Drug Use and Health (NSDUH; see Section 2.2), the mortality data were from the CDC Wonder Cause of Death database (see Section 2.3), and the socio-demographic variables were from U.S. Census (see Section 2.4).

### 2.2. State-Level Prevalence Estimates

The prevalence of opioid dependence in each state was based on estimates from adults in the National Survey on Drug Use and Health (NSDUH).

The NSDUH is a nationally representative study that collects data on substance use and dependence. The NSDUH interviews approximately 70,000 persons aged 12 years and above every year. This study used survey data available from 2004–2014 in the analysis. It samples residents of households, non-institutional group quarters, and civilians living on military bases. Respondents were selected through a stratified multistage area probability sample, representing each of the 50 states and the District of Columbia. Household addresses were randomly selected, then 1–2 members of the household may be selected to complete a full interview. Participation is voluntary and the participants are reimbursed with USD 30 for their contributions to the survey. The participants conduct the interviews in their own home. Participants answer the questions in private and the interview takes, on average, one hour to complete. All the information collected is confidential and used only for statistical purposes.

Statistical imputation is used to replace missing or ambiguous values after editing for key variables. The procedure applies probabilistic statistical methods to identify another similar respondent whose data were used to replace the missing values or ambiguous responses in the recipient record. Methodological documentations of the NSDUH are published online by SAMHSA. Analysis weights are created so that estimates are representative of the target population. Response rates for each year of the study were ranged from 82–91% (Appendix B, Table A1).

Geographic identifiers are collected in the NSDUH, but micro-data on opioid dependence by states are not distributed in the public domain. State-level data on past-year opioid dependence were obtained from the Substance Abuse and Mental Health Services Administration (SAMHSA) through a special data request that approved the use of and provided data from 2005–2014. Year-by-year data by state was not provided by SAMHSA due to small numbers after stratification by age and sex to protect the confidentiality of the participants. Therefore, the state-level was provided as the weighted mean of the 10-year sample. That is, analysis weights were applied to the estimates to be representative of the target population. Then, data over the 10 years were pooled by (by calculating the mean prevalence estimates) SAMHSA across the years, to provide state-level prevalence estimate for our state-level associates analyses.

Past-year opioid dependence, stratified by age, sex, and state, was defined according to the 4th edition of the Diagnostic and Statistical Manual of Mental Disorders (DSM-IV). Included in the interview were questions based on the six criteria in the DSM-IV for diagnosing substance-use dependence. For heroin and pain relievers, a seventh withdrawal criterion was added and was defined by a respondent reporting having experienced a number of withdrawal symptoms (e.g., having trouble sleeping, cramps, and hand tremble). A respondent was defined as having dependence if they met three or more of the seven dependence criteria for heroin or opioid use. Three categories of opioid dependence were examined: (1) heroin dependence, (2) NMPO dependence, and (3) any opioid dependence (including those who met either of the first two categories). Categories 1 and 2 are not mutually exclusive; they include people who met criteria for both heroin and NMPOs.

### 2.3. Opioid-Overdose Death Rates

Mortality caused by overdose (death codes: X40–X44, X60-64, X85, Y10-Y14) due to opioids (T40.0-T40.4, T40.6) was retrieved from the CDC Wonder Cause of Death database. These codes were selected because they represent opioid-related deaths. A description of the codes can be found in Appendix C (Table A2). Death rates by each of the U.S. states between 2005–2014 were extracted.

### 2.4. State-Level Socio-Demographic Characteristics

The state-level socio-demographic characteristics examined included: GDP/capita, mean personal income, high-school completion rate, rurality, and unemployment rate. GDP/capita was calculated as each state’s annual Gross Domestic Product divided by their resident population [18]. Personal income was defined as the annual sum of all employment-related or investment-related earnings received by an individual [18]. High-school completion rate was defined as the proportion of each state’s adult population with a high-school diploma or higher education level [19]. Rurality was defined as the proportion of each state’s population living in a community of under 2500 residents [20,25]. Unemployment rate was defined as the proportion of all individuals aged 16+ who were not employed at the time, despite actively searching for work [26]. As the U.S. Census is a decennial survey, rurality data were collected from 2010; all other characteristics were averaged over the years 2005–2014 to correspond with the state-level opioid dependence data.

### 2.5. Statistical Analysis

Firstly, we examined the state-level variations in opioid dependence by presenting the prevalence estimates by the states. All analyses were stratified by the three opioid dependence categories. We presented the weighted prevalence overall, and by age and sex (data available in Appendix A). As mentioned in the data source section above, the prevalence estimates were calculated as the mean over the 10 years of our study period, due to lack of year-by-year data availability.

Second, we examined the correlation of opioid dependence with opioid-related death rates. Pearson’s correlations were used to examine the associations between state-level prevalence estimates with state-level opioid-related mortality. Analyses were stratified by the three opioid dependence categories, and by age and sex groups. The associations were presented in scatterplots.

Thirdly, Pearson’s correlations were conducted to examine the correlations between state-level socio-demographic factors and state-level prevalence of opioid dependence. Correlations were calculated separately for each of the three opioid dependence categories. Data visualizations were created in Tableau and correlation analyses were conducted in StatsNotebook 0.1.0, which is an open source statistical package built on R.

## 3. Results

### 3.1. State-Level Variations

The prevalence of heroin dependence ranged from <0.1% to 0.4% (mean = 0.1, SD = 0.1), and NMPO dependence ranged from 0.2–1.2% (mean = 0.6, SD = 0.2; see Appendix A). Pooled state-level data across 2005–2014 showed that Connecticut, the District of Columbia, and New Jersey had the highest prevalence of heroin dependence, followed by Delaware, Oregon, Massachusetts, Pennsylvania, and Rhode Island (see Appendix A). The states with the highest rates of NMPO dependence were Kentucky (1.2%) and West Virginia (1.2%), followed by New Hampshire and Tennessee at 0.9% and Arizona, Louisiana, Ohio, and Oklahoma at 0.8%. These geographical variations were similar across age groups and sexes (see Appendix A).

### 3.2. Correlation with Opioid-Related Death Rates

Across all the states, opioid-related death rates ranged from 3.2 to 31.6 per 100 thousand (mean = 10.4, SD = 5.7). The rates were lowest in Nebraska (3.2), Hawaii (3.9), and Mississippi (3.9), and highest in West Virginia (31.6), New Hampshire (23.4), and New Mexico (see Appendix A).

There was a positive association between state-level opioid dependence rates and state-level opioid-related death rates (r = 0.69, *p* < 0.001; see Figure 1). An interactive data visualization tool is available online (https://public.tableau.com/app/profile/janni.leung/viz/Opioid-relateddeathratesbystate-levelprevalenceofopioidheroinandnon-medicalprescriptionopioiddependenceintheUSA-LeungJ2022/Dashboard (accessed on 15 December 2021)). Opioid-related mortality was more strongly associated with NMPO (r = 0.62, *p* < 0.001) than with heroin dependence (r = 0.45, *p* = 0.001. Both heroin (r = 0.52, *p* < 0.001) and NMPO-dependence (r = 0.64, *p* < 0.001) rates were significantly correlated with opioid death rates in the 18–25 years age group. In the 26+ years age groups, NMUPO-dependence prevalence was significantly associated with opioid death rates, but heroin dependence was not.

### 3.3. Socio-Demographic Correlates

Descriptive statistics for the socio-demographic factors across the states are available in Appendix A. Higher unemployment (r = 0.38, *p* = 0.006) and lower high school completion (r = −0.33, *p* = 0.018) rates were significantly associated with a higher prevalence of state-level opioid dependence (see Table 1). At a state level, a lower GDP/capita (r = −0.39, *p* = 0.005), lower personal income (r = −0.42, *p* = 0.002), and lower high school completion rate (r = −0.28, *p* = 0.007) were associated with NMPO dependence prevalence. In contrast, higher GDP/capita (r = 0.48, *p* < 0.001), higher personal income (r = 0.59, *p* < 0.001), and lower proportion living in rural areas (r = −0.40, *p* = 0.004), were associated with a prevalence of heroin dependence in the states.

## 4. Discussion

This study examined the associations between NMPO and heroin dependence with state-level opioid-related deaths and socio-demographic characteristics in the United States. The prevalence of heroin dependence was associated with opioid-related deaths in young-to-middle-aged groups, possibly because of the low prevalence in adults over 50 years of age. Prevalence of NMPO dependence was correlated with opioid-related deaths in adults from all age groups. We found that indicators of higher socioeconomic advantage were associated with a higher prevalence of heroin dependence, but negatively associated with NMPO dependence. This finding may be explained by the greater accessibility of heroin in bigger cities, which are primarily located in higher-income states [17]. In contrast, the prevalence of NMPO dependence was highest in areas of lower educational and financial attainment, including the states of Kentucky, West Virginia, and New Hampshire. Given the descriptive nature of the study, no causal interpretation is possible.

The finding that the prevalence of opioid dependence was significantly correlated with opioid-related mortality, corroborated with the literature that determined that individuals with long-term opioid dependence are at a higher risk of overdose [3,14]. Our finding that heroin dependence was significantly associated with opioid-related deaths in the young-to-middle-aged groups imply that younger adults comprise the primary demographic for heroin use, so they may constitute a candidate for targeted awareness and intervention initiatives [27]. While heroin use had traditionally been a predominately male problem, we found that the prevalence of heroin and NMPO dependence were associated with opioid-related deaths in females. The needs of people who use drugs are likely to differ by sex. For example, females who use NMPOs are more likely to have comorbid depression [28]. Our findings provide further support for gender-appropriate strategies to reduce opioid-related harms.

The finding that NMPO dependence was correlated with opioid-related deaths in middle-aged and older adults (aged 26+ years) is relevant for public health strategies to address the opioid epidemic. Compared to younger people, middle-aged and older adults are more likely to develop dependence initiated by prescribed pain relievers [29], often on higher-potency analgesics [30]. Older patients experience greater levels of pain and polypharmacy due to other chronic conditions may be more vulnerable to opioid-drug (e.g., concomitant benzodiazepine use) and opioid-disease interactions (e.g., exacerbation of impaired respiratory or renal function) [31].

Previous studies observed that the factors that influence the association between opioid use and state-level socio-demographics include the relatively expensive (albeit decreasing) street price of heroin [17], as well as its greater accessibility in major cities [32]. In contrast, NMPO dependence is most prominent in less economically and educationally advantaged states, including Kentucky and New Hampshire, showing an association between NMPOs and lower education [33,34]. A recent study examined county-level labor market outcomes in ten states using the U.S.’s Prescription Drug Monitoring Programs and labor data from the Bureau of Labor Statistics [35]. It reported a negative effect on employment and labor force participation rates when measuring the number of opioids prescriptions written and the number of doses prescribed. However, the authors cautioned against restricting the supply of prescription opioids as a strategy to boost employment and labor force participation. Imposing barriers to access opioids could lead to unintentional consequences, including leaving those in need without effective pain management, and people who use pharmaceutical opioid switching to other forms of harmful opioids [4].

Our findings imply that public health interventions for the opioid epidemic need to consider the broad variety of socio-demographic groups affected. There is strong evidence that opioid agonist treatment (OAT), for which coverage is currently low, is highly effective in reducing opioid use and opioid-related deaths, including overdose, suicide, HIV, hepatitis C virus, and injuries [36]. A recent study demonstrated that scaling up the use of OATs has the potential to avert 7.7% of deaths in Kentucky [36]. Another intervention that could be effective to reduce opioid-related deaths would be to increase the provision of naloxone. By increasing the awareness of the co-prescription of naloxone with opioids, it might be a sensible intervention, as opioid dependence increases. Increased access to effective interventions for all affected by opioids is a clear public health priority in the U.S., and also countries in which an increase in opioid-related harms is emerging [37,38].

This study has a few limitations. The NSDUH findings cannot be generalized to special and high-risk populations, such as incarcerated individuals or intravenous drug users, among whom mortality risk due to overdose is much higher.

Another limitation is that our state-level measures were pooled across multiple years, precluding an analysis of temporal trends by state. We cannot assume that our variables have not changed across time, so our findings cannot be used to inform population-level changes over time. The validity of year-by-year variations in the prevalence of heroin use in general population surveys has been under debate [39]. Under-reporting is a major limitations of population surveys on substance use behaviors. The resulting low number of cases may lead to year-by-year variations that may not reflect the true prevalence and combining multiple years of data was proposed to provide a more stable population estimate. Further, the NSDUH made changes to items asking about opioid use since 2015, therefore data since then may not be directly comparable to the previous data. Our current analysis is reasonable because it used multiple years of data combined before the change of method of measuring opioid use. Future studies that are enabled when new data becomes available will provide epidemiological information on any changes in regional and socio-demographic groups affected.

We did not have data that explicitly distinguishes synthetic opioids, nor illicitly manufactured synthetic opioids, which drove opioid mortality in the U.S.A. [13]. An analysis on a sub-sample of young adults who reported past-month NMPO use from the Rhode Island Young Adult Prescription Drug Study found that 1 in 10 was exposed to fentanyl-contaminated heroin, with 60% reporting that they were unaware of the contamination before use [40]. Future epidemiological studies should consider the changing landscape of opioid use in the United States, with recent intentional and unintentional increased uptake in highly potent illicitly manufactured fentanyl and other novel psychoactive substances [41].

Future studies should examine the concurrent use of opioids with other substances to inform interventions, because there are differences in the use of other substances among people who use NMPOs [42]. A recent study of U.S. adults using prescription opioids found that those who misused both their own prescriptions and opioids prescribed to others were more likely to be co-using cannabis, benzodiazepine, and heroin, compared to those who were only misusing their own prescriptions [43]. They were also more likely to meet criteria for an opioid use disorder. In people engaging in NMPOs with a prescription, extending substance use beyond their own prescriptions puts them at higher risk of negative mental and physical consequences.

In addition to understanding the associates of an opioid-related burden, our findings have other public health implications, (e.g., elevated rates of hepatitis C have been observed in states with higher rates of intravenous OxyContin misuse) [44]. Future studies should examine the concurrent use of opioids with other substances. Looking at opioids alone provides an incomplete picture because concurrent use of respiratory depressants, such as alcohol and benzodiazepines, increases the risk of fatal overdoses. Although opioid dependence is mainly an adult disorder, we need evidence-based preventive strategies earlier in life, because NMPO use in late-childhood and adolescence is a strong risk factor for transitioning to heroin use in early adulthood [45]. We also need to consider the changing epidemiology of opioid use in the United States, with a recent uptake in the use of high-potency synthetic opioids [46].

## 5. Conclusions

The prevalence of opioid dependence was not only associated with mortality in young adults, but also associated with opioid-related mortality in older adults by extra-medical use of pharmaceutical opioids. Prevalence rates of opioid dependence were associated with a range of geographical and socio-demographic groups, including populations with high income and education levels. Public health interventions and public policies should take a broader view of the variety of populations affected by opioid misuse, inclusive of people affected by both heroin and prescription opioid use, the age ranges, and genders affected. We need to be increasing the access to effective interventions for all, to reduce opioid-related disease burden in the U.S.A.

## Figures and Tables

**Figure 1 ijerph-19-03825-f001:**
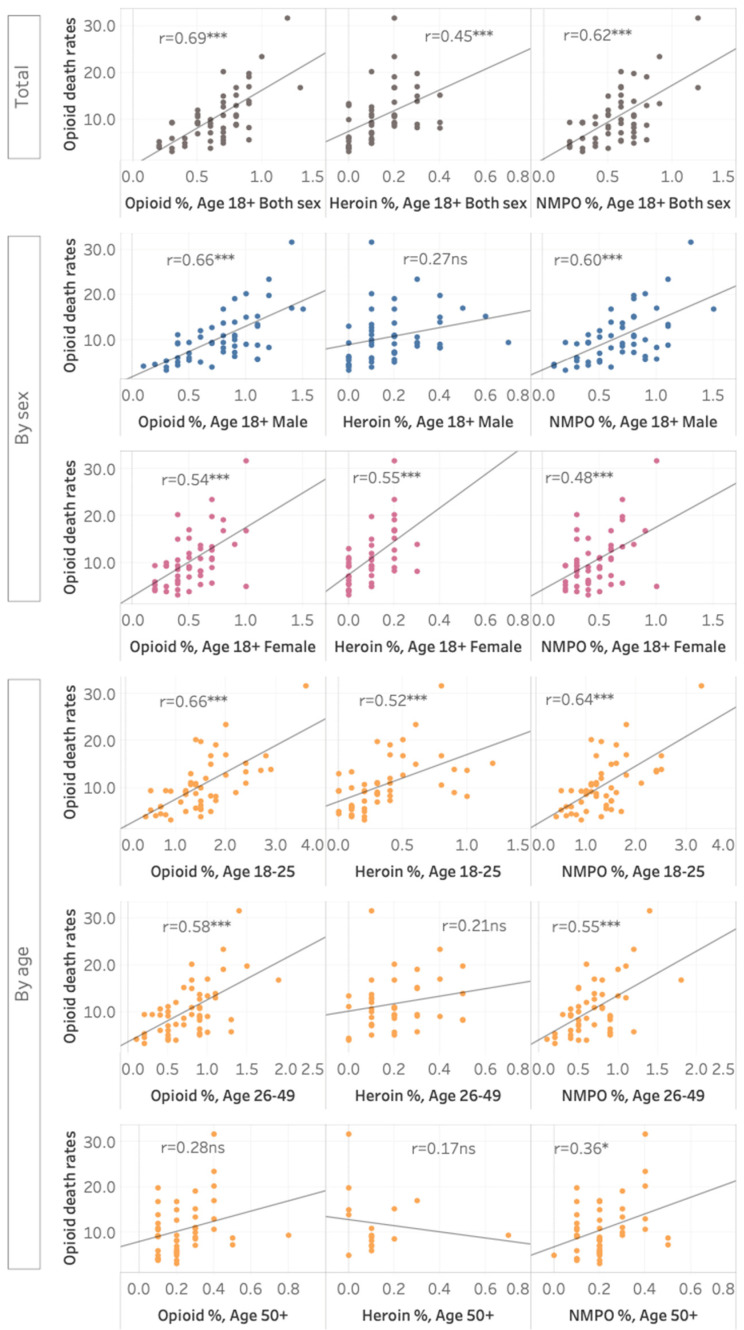
State-level prevalence of opioids dependence, heroin dependence, and NMPO dependence. An interactive data visualization tool is available online (https://public.tableau.com/app/profile/janni.leung/viz/Opioid-relateddeathratesbystate-levelprevalenceofopioidheroinandnon-medicalprescriptionopioiddependenceintheUSA-LeungJ2022/Dashboard (accessed on 15 December 2021)). * *p* < 0.05, *** *p* < 0.001.

**Table 1 ijerph-19-03825-t001:** Correlations between state-level socio-demographic factors and state-level prevalence of dependence.

State-Level Socio-Demographic Factors	Correlation with State-Level Past Year DSM-IV Prevalence ^f^					
	Opioid Dependence		Heroin Dependence		NMPO Dependence	
	r	*p*	r	*p*	r	*p*
GDP/capita ^a^	−0.20	0.170	0.48	<0.001	−0.39	0.005
Mean personal income ^b^	−0.22	0.122	0.59	<0.001	−0.42	0.002
High-school completion ^c^	−0.33	0.018	0.19	0.189	−0.38	0.007
Rurality ^d^	0.03	0.850	−0.40	0.004	0.18	0.198
Unemployment rate ^e^	0.38	0.006	0.27	0.059	0.29	0.036

^a^ Average GDP/capita during 2005–2014; ^b^ average mean personal income over 2005–2014; ^c^ average % of residents who completed high school during 2005–2014; ^d^ % rural in 2010; (rural defined as city/town with <2500 pop); ^e^ average unemployment rate during 2005–2014; ^f^ pooled prevalence of dependence across 2005–2014 of each state; NMPOs: non-medical prescription opioids.

## Data Availability

The data presented in this study are available in the Appendix A.

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
