# Peer review of "State-Level Prevalence and Associates of Opioid Dependence in the USA"

_ijerph, 2022, doi:10.3390/ijerph19073825_

Round 1

Reviewer 1 Report

Minor comments:

- The aim of the manuscript is missing in the abstract

- I think something is missing at Line 21: their associations [with?] state-level socio-demographic characteristics

Major comments:

Section 2: Materials and Methods

This section needs a thorough review by providing more details, as it does not allow to understand how the authors treated their data and for the socio-demographic variables (if I am not mistaken, see the related point below) the rationale for including them in the analysis. 

In particular:

- Section 2.2: State-level Prevalence

Apart from the study design, no information is provided about this survey (e.g. what is the data collection method used? personal interviews, online surveys, paper questionnaires etc. Is the survey anonymous?). These information are very relevant, especially when socially undesirable behaviours are analysed.

The authors refer to the variable used as “Prevalence of opioid dependence”. However usually the term  “dependence” is used when there is a clinical diagnosis behind. If this is not the case, this term should not be used. The authors write this was defined according  to the 4th edition of the Diagnostic and Statistical Manual of Mental Disorders (DSM-IV). Does that mean that the questionnaire contain a screening tool? These set of information should be provided.

Line 80: The method used for the statistical imputation applied to replace missing or ambiguous values should be better described.

Lines 90-92: “the state-level was provided as the weighted mean of the 10-year sample. That is, data were pooled by SAMHSA across the years to provide state-level prevalence estimate for our state-level associates analyses.”

This would need to be explained in details as it is not clear how these data were treated. E.g. weighted (for what?) mean, .. pooled (how?)..

- Section 2.3: Opioid Dependence Categories

 If I well understand this is how the information described in 2.2. State-level Prevalence were treated, isn’t it? If this is the case then this section should be part of section 2.2. because like this is seems something different.

- Section 2.4: Opioid Overdose Death Rates

Lines 101-102: Mortality caused by overdose (death codes: X40-X44, X60-64, X85, Y10-Y14) due to 101 opioids (T40.0-T40.4, T40.6). The codes provided are not useful for the reader unless they are explained. A brief explanation is needed here.

- Section 2.5: State-level Socio-demographic characteristics

Lines 114-116: What does this mean “all other characteristics were averaged over the years 2005-2014”? The thing I cannot understand from this explanation and in the absence of descriptive statistics (it is perfectly understandable that these have not been provided given that 50 states are analysed) is if the authors are assuming that these variables have not changed over time. This would be difficult to justify, especially because the analysis covers the years of the economic crisis and constitute an enormous limitation.

Section 4: Discussion:

Lines 170-171: “Given the ecological nature and period of data analysed of this study, caution is warranted in a causal interpretation of these findings.” I think that the analysis provided is appreciable but really simple. I would modify this sentence by saying that given the descriptive nature of the study no causal interpretation is possible.

Lines 251-252 “In addition to reducing opioid-related burden, our findings have other public health implications”. Put like this the sentence seems to say that the findings of the study have has implication a reduction in opioid-related burden…

Section 5: Conclusions

This conclusion “Public health interventions and public policies should take a broader view of the variety of populations affected by opioid misuse, increasing access to effective interventions for all, to reduce opioid-related disease burden.” sounds  a bit simplistic. I am referring to the fact that the authors include in their analysis three categories of Opioid Dependence and a set of socio-demographic variables, different age ranges, and they take into account of gender, eventually obtaining a wide set of results. However the conclusions are quite generic, to the point that they could be applied to almost all substances and in all countries. I would advise to make them more targeted.

Author Response

We would like to thank you for taking the time to read and provide comments for our manuscript. We found the review was very helpful for improving our paper.

Minor comments:

The aim of the manuscript is missing in the abstract

            [Response]: We included the aim in the abstract, as follows:

We aim to examine state-level prevalence of heroin and NMPO dependence and their associations with opioid-related mortality and state-level socio-demographic profiles.

I think something is missing at Line 21: their associations [with?] state-level socio-demographic characteristics

[Response]: Thank you and apologies, with is correct, we have now added “with” back into the sentence.

Major comments:

Section 2: Materials and Methods

This section needs a thorough review by providing more details, as it does not allow to understand how the authors treated their data and for the socio-demographic variables (if I am not mistaken, see the related point below) the rationale for including them in the analysis. In particular: Section 2.2: State-level Prevalence

Apart from the study design, no information is provided about this survey (e.g. what is the data collection method used? personal interviews, online surveys, paper questionnaires etc. Is the survey anonymous?). These information are very relevant, especially when socially undesirable behaviours are analysed.

[Response]: We have revised our manuscript to provide information about this survey, as follows:

“The NSDUH is a nationally representative study that collects data on substance use and dependence. The NSDUH interviews approximately 70,000 persons aged 12 and above every year. This study used survey data available from 2004-2014 in the analysis. It samples residents of households, non-institutional group quarters and civilians living on military bases. Respondents were selected through a stratified multistage area probability sample, representing each of the 50 states and the District of Columbia. Household addresses are randomly selected, then 1-2 members of the household may be selected to complete a full interview. Participation is voluntary and the participants are reimbursed with $30 for their contributions to the survey. The participants do the interviews in their own home. Participants answer the questions in private and the interview takes on average one hour to complete. All information collected is confidential and used only for statistical purposes.

Statistical imputation is used to replace missing or ambiguous values after editing for key variables. Analysis weights are created so that estimates will be representative of the target population. Response rates for each year of the study were ranged from 82%-91%”.

The authors refer to the variable used as “Prevalence of opioid dependence”. However usually the term  “dependence” is used when there is a clinical diagnosis behind. If this is not the case, this term should not be used. The authors write this was defined according  to the 4th edition of the Diagnostic and Statistical Manual of Mental Disorders (DSM-IV). Does that mean that the questionnaire contain a screening tool? These set of information should be provided.

[Response]: The questionnaire involved a screening tool where the participants were classified as having opioid/NMPO/heroin dependence based on their responses to the DSM-IV diagnostic criteria. If a respondent were to disclose past year substance use in the core drug module or the noncore special drugs module, they would be asked the DSM-IV diagnostic criteria for substance dependence. If the respondent were to meet three or more out of the seven criterions, they would be categorised as having dependence.

We have revised our manuscript to make it more clear for the reader as to how the respondents were categorised as having opioid/NMPO/heroin dependence:

“Past-year opioid dependence, stratified by age, sex, and state was defined according to the 4th edition of the Diagnostic and Statistical Manual of Mental Disorders (DSM-IV). Included in the interview were questions based on the six criteria in the DSM-IV for diagnosing substance use dependence. For heroin and pain relievers, a seventh withdrawal criterion was added and was defined by a respondent reporting having experienced a number of withdrawal symptoms (e.g., having trouble sleeping, cramps, hand tremble). A respondent was defined as having dependence if they met three or more of the seven dependence criteria for heroin or opioid use. Three categories of opioid dependence were examined: 1) heroin dependence, 2) NMPO dependence, and 3) any opioid dependence (including those who met either of the first two categories). Categories 1 & 2 are not mutually exclusive; they include people who met criteria for both heroin and NMPO.”

Line 80: The method used for the statistical imputation applied to replace missing or ambiguous values should be better described.

[Response]: We have revised our manuscript to provide information on the statistical imputation method used for missing or ambiguous values, as follows:

“Statistical imputation is used to replace missing or ambiguous values after editing for key variables. The procedure applies probabilistic statistical methods to identify another similar respondent whose data were used to replace the missing values or ambiguous responses in the recipient record. Methodological documentations of the NSDUH are published online by SAMHSA.”

Lines 90-92: “the state-level was provided as the weighted mean of the 10-year sample. That is, data were pooled by SAMHSA across the years to provide state-level prevalence estimate for our state-level associates analyses.” This would need to be explained in details as it is not clear how these data were treated. E.g. weighted (for what?) mean, .. pooled (how?)..

[Response]: We agree and have added the additional details, as follows,

“Therefore, the state-level was provided as the weighted mean of the 10-year sample. That is, analysis weights were applied to the estimates to be representative of the target population. Then, data over the 10 years were pooled by (by calculating the mean prevalence estimates) SAMHSA across the years, to provide state-level prevalence estimate for our state-level associates analyses.”

- Section 2.3: Opioid Dependence Categories

 If I well understand this is how the information described in 2.2. State-level Prevalence were treated, isn’t it? If this is the case then this section should be part of section 2.2. because like this is seems something different.

[Response]: We have revised this section and decided to merge section 2.3 with 2.2.

- Section 2.4: Opioid Overdose Death Rates

Lines 101-102: Mortality caused by overdose (death codes: X40-X44, X60-64, X85, Y10-Y14) due to 101 opioids (T40.0-T40.4, T40.6). The codes provided are not useful for the reader unless they are explained. A brief explanation is needed here.

            [Response]: We added a brief explanation that these codes were selected because they represent opioids-related deaths. In addition, we have now provided a table in the appendix for the reader to gain a better understanding on what the death codes specifically mean. The table is as follows:

Appendix B

Table 1ab. Description of opioid-related ICD-10 death codes

Drug poisoning intent

ICD-10 codes

Unintentional

X40-X44

 Suicide/self-harm

X60-X64

Assault by drug poisoning

X85

 Undetermined drug poisoning Intent

Y10-Y14

Contributing drug of overdose

ICD-10 codes

Opium

T40.0

Heroin

T40.1

Natural and semisynthetic opioids

T40.2

Methadone

T40.3

Synthetic opioids, other than methadone

T40.4

Other unspecified narcotics

T40.6

- Section 2.5: State-level Socio-demographic characteristics

Lines 114-116: What does this mean “all other characteristics were averaged over the years 2005-2014”? The thing I cannot understand from this explanation and in the absence of descriptive statistics (it is perfectly understandable that these have not been provided given that 50 states are analysed) is if the authors are assuming that these variables have not changed over time. This would be difficult to justify, especially because the analysis covers the years of the economic crisis and constitute an enormous limitation.

[Response]: The reviewer is right that year-by-year descriptive statistics by the 50 states have not been provided. The reviewer make a good point about assumptions on changes overtime. We cannot assume that the variables have not changed over time. In the absence of data, we discussed this in our limitation section, as follows:

“Another limitation is that our state-level measures were pooled across multiple years, precluding an analysis of temporal trends by state. We cannot assume that our variables have not changed across time, so our findings cannot be used to inform population-level changes over time. The validity of year-by-year variations in the prevalence of heroin use in general population surveys has been under debate [39]. Under-reporting is a major limitations of population surveys on substance use behaviours. The resulting low number of cases may lead to year-by-year variations that may not reflect the true prevalence, and combining multiple years of data had been proposed to provide a more stable population estimate.”

Section 4: Discussion:

Lines 170-171: “Given the ecological nature and period of data analysed of this study, caution is warranted in a causal interpretation of these findings.” I think that the analysis provided is appreciable but really simple. I would modify this sentence by saying that given the descriptive nature of the study no causal interpretation is possible.

[Response]: We agree and have revised our sentence as suggested.

Lines 251-252 “In addition to reducing opioid-related burden, our findings have other public health implications”. Put like this the sentence seems to say that the findings of the study have has implication a reduction in opioid-related burden…

[Response]: We agree have revised it, as follows:

“In addition to understanding the associates of opioid-related burden, our findings have other public health implications.”

Section 5: Conclusions

This conclusion “Public health interventions and public policies should take a broader view of the variety of populations affected by opioid misuse, increasing access to effective interventions for all, to reduce opioid-related disease burden.” sounds a bit simplistic. I am referring to the fact that the authors include in their analysis three categories of Opioid Dependence and a set of socio-demographic variables, different age ranges, and they take into account of gender, eventually obtaining a wide set of results. However the conclusions are quite generic, to the point that they could be applied to almost all substances and in all countries. I would advise to make them more targeted.

[Response]: We thank the reviewer for this suggestion and have revised our conclusion, as follows:

Public health interventions and public policies should take a broader view of the variety of populations affected by opioid misuse, inclusive of people affected by both heroin and prescription opioids use, the age ranges, and genders affected. We need to be increasing access to effective interventions for all, to reduce opioid-related disease burden in the USA.”

Reviewer 2 Report

This study aims to examine state-level prevalence of 16 heroin and NMPO dependence and their associations with opioid-related mortality and state-level 17 socio-demographic profiles. Even if it comes with strong limitations, especially on the point of view of a lacking in yearly data availability, so making the analysis kind of poor from the statistical robustness point of view, it still can be a strong exploratory analysis and an interesting starting point for further research. Nonetheless, in my opinion the second section about "Matherial and Methods" is poor, confused and not self-explanatory, requiring the reader to move up and down through the paper in order to link the estimation with the defective information given. I suggest rewriting the second section almost completely, following the following points, if they can be useful.

1) The paragraph 2.1 is extremely insufficient. Please, clarify much more extensively what do you mean by "a cross-sectional analysis of state-level prevalence, mortality, and socio-demographic variables between 2005-2014."

2) Please, check coherence between tables' labels. As far as I can see, at line 132 you refer to a non-existing table S2. In the Supplementary materials, tables are listed as S1a, S1b and S1c.

3) Moreover, both the tables in the text are labeled as Table 1. Even if one is in the main text and the other in the appendix, they should not have the same label.

4) In my opinion, the paragraph about Statistical Analysis has to be much more extensive and clear. I struggled to figure out the way you were analyzing data, and the only way to help myself was to move forward in a deep study of figure 1, then coming back to paragraph 2.6 in order to finally understand the underlying technique.

5) Another thing which is confusing for me, is that data are explained within four paragraphs which follow the study design subsection and anticipate the statistical analysis subsection. I think that paragraphs 2-5 are materials, while paragraphs 1 and 6 are methods. Please, take into consideration the possibility of restructuring the second section accordingly.

6) Line 88: "Year-by-year data by state was not provided due to small numbers after stratification by age and sex to protect confidentiality of the participants." This sentence is slightly unclear. Please, clarify if the decision of not providing year-by-year data is up to you or to SAHMSA. If the latter is the case, then on the one side the reader can understand that this choice was unavoidable, on the other side authors should state much more extensively all the limitations this issue brings with it.

7) Again, I can understand that you cannot provide descriptive statistics for all the states, but you still need to provide a table with at least means and standard deviations for the variables between countries. The readers need some benchmarks to understand in the framework what a high/low/average mean of opioids, or GDP per capita or high school completion can be and which ones are the variables with the highest variabilities in the comparison of countries. 

Author Response

We would like to thank you for taking the time to read and provide comments for our manuscript. We found the review was very helpful for improving our paper.

This study aims to examine state-level prevalence of 16 heroin and NMPO dependence and their associations with opioid-related mortality and state-level 17 socio-demographic profiles. Even if it comes with strong limitations, especially on the point of view of a lacking in yearly data availability, so making the analysis kind of poor from the statistical robustness point of view, it still can be a strong exploratory analysis and an interesting starting point for further research. Nonetheless, in my opinion the second section about "Matherial and Methods" is poor, confused and not self-explanatory, requiring the reader to move up and down through the paper in order to link the estimation with the defective information given. I suggest rewriting the second section almost completely, following the following points, if they can be useful.

[Response]: We thank the reviewer for their comments and have revised our paper as best we can for improvements.

1) The paragraph 2.1 is extremely insufficient. Please, clarify much more extensively what do you mean by "a cross-sectional analysis of state-level prevalence, mortality, and socio-demographic variables between 2005-2014."

[Response]: Our prevalence, mortality, and socio-demographic variables were from different data source. We clarified this in our revised paper and refer readers to the additional details in their corresponding sections, as follows:

“Three data sources were used for this study with information presented in the below corresponding sections. The state-level prevalence estimates were sourced from the National Survey on Drug Use and Health (NSDUH; see 2.2), the mortality data were from the CDC Wonder Cause of Death database (see 2.3), and the socio-demographic variables were from U.S. Census (see 2.4).”

2) Please, check coherence between tables' labels. As far as I can see, at line 132 you refer to a non-existing table S2. In the Supplementary materials, tables are listed as S1a, S1b and S1c.

[Response]: We would like to thank the reviewers for indicating that we check the table labels to ensure they are coherent. We have now checked the table labels to ensure they match the label given in the supplementary material.

3) Moreover, both the tables in the text are labeled as Table 1. Even if one is in the main text and the other in the appendix, they should not have the same label.

[Response]: We have now revised the appendix labels to make it clearer about the tables we were referring to. The appendix labels are now Table 1aa and Table 1ab.

4) In my opinion, the paragraph about Statistical Analysis has to be much more extensive and clear. I struggled to figure out the way you were analyzing data, and the only way to help myself was to move forward in a deep study of figure 1, then coming back to paragraph 2.6 in order to finally understand the underlying technique.

[Response]: We revised our statistical analysis to provide more details, as follows,

“Firstly, we examined state-level variations in opioid dependence by presenting the prevalence estimates by the states. All analyses were stratified by the three opioid dependence categories. We presented the weighted prevalence overall, and by age and sex (data available in Table S1a). As mentioned in the data source section above, the prevalence estimates were calculated as the mean over the 10 years of our study period, due to lack of year-by-year data availability.

Second, we examined the correlation of opioid dependence with opioid-related death rates. Pearson's correlations were used to examine the associations between state-level prevalence estimates with state-level opioid-related mortality. Analyses were stratified by the three opioid dependence categories, and by age and sex groups. The associations were presented in scatterplots.

Thirdly, Pearson’s correlations were conducted to examine the correlations between state-level socio-demographic factors and state-level prevalence of opioid dependence. Correlations were calculated separately for each of the three opioid dependence categories. Data visualization were created in Tableau and correlation analyses were conducted in R.”

5) Another thing which is confusing for me, is that data are explained within four paragraphs which follow the study design subsection and anticipate the statistical analysis subsection. I think that paragraphs 2-5 are materials, while paragraphs 1 and 6 are methods. Please, take into consideration the possibility of restructuring the second section accordingly.

[Response]: We would like to thank the reviewer for bringing this to our attention. We now see how this section could be confusing for the reader and have decided to merge section 2.2 and 2.3 together. In addition, we would like to keep paragraph 1 and 6 separated as paragraph 2.1 is the study design and section 2.5 is the statistical analysis. Both can be a stand-alone section and we believe having the study design at the beginning of the material and methods section allows for a sound introduction of the section.

6) Line 88: "Year-by-year data by state was not provided due to small numbers after stratification by age and sex to protect confidentiality of the participants." This sentence is slightly unclear. Please, clarify if the decision of not providing year-by-year data is up to you or to SAHMSA. If the latter is the case, then on the one side the reader can understand that this choice was unavoidable, on the other side authors should state much more extensively all the limitations this issue brings with it.

[Response]: The decision to not provide year-by-year data was that of SAMHSA. We have now revised our manuscript to provide better clarification, as follows:

“Year-by-year data by state was not provided by SAMHSA due to small numbers after stratification by age and sex to protect confidentiality of the participants.”

7) Again, I can understand that you cannot provide descriptive statistics for all the states, but you still need to provide a table with at least means and standard deviations for the variables between countries. The readers need some benchmarks to understand in the framework what a high/low/average mean of opioids, or GDP per capita or high school completion can be and which ones are the variables with the highest variabilities in the comparison of countries. 

[Response]: We agree that descriptive statistics would be useful and added these into our revised paper, as follows,

“Prevalence of heroin dependence ranged from <0.1% to 0.4% (mean=0.1, SD=0.1), and NMPO dependence ranged from 0.2%-1.2% (mean=0.6, SD=0.2; see Table S2a).”

“Across all the states, opioid-related death rates ranged from 3.2 to 31.6 per 100 thou-sand (mean=10.4, SD=5.7). The rates were lowest in Nebraska (3.2), Hawaii (3.9) and Mississippi (3.9), and highest in West Virginia (31.6), New Hampshire (23.4), and New Mexico (see Table S2a).”

“Descriptive statistics for the socio-demographic factors across the states are available in Table S2b.”

Table S2a. Descriptive statistics across states for opioids variables

Mean

SD

Median

Range

Min

Max

Opioid death rates*

10.4

5.7

9.3

28.4

3.2

31.6

Opioid dependence

0.6

0.2

0.7

1.1

0.2

1.3

Heroin dependence

0.1

0.1

0.1

0.4

<0.1

0.4

NMPO dependence

0.6

0.2

0.6

1.0

0.2

1.2

*per 100,000

Table S2b. Descriptive statistics across states for socio-demographic factors

Mean

SD

Median

Range

Min

Max

GDP/capita a

     49,168

     18,564

     46,458

   133,140

     31,741

   164,881

Mean personal income b

     40,708

       6,869

     39,384

     31,707

     30,678

     62,384

High school completion c

87.9

3.1

88.8

10.9

81.5

92.4

Rurality d

0.3

0.2

0.3

0.8

0.0

0.8

Unemployment rate e

6.4

1.3

6.5

5.8

3.3

9.1

a Average GDP/capita over 2005-2014; b Average mean personal income over 2005-2014; c Average % of residents who completed high school over 2005-2014; d % rural in 2010; (rural defined as city/town with <2,500 pop); e Average unemployment rate over 2005-2014; f Pooled prevalence of dependence across 2005-2014 of each state; NMPO: non-medical prescription opioids.

Round 2

Reviewer 2 Report

The second paragraph was improved sufficiently and those parts added to the other paragraphs are useful for the readers. Now, I think the quality is acceptable.

Author Response

We would like to thank the reviewer for taking the time to provide feedback throughout the review process. We believe the comments and feedback they provided helped us improve our paper.

Yours sincerely,  

Janni Leung and collaborators